# Investigating Conceptual Blending of a Diffusion Model for Improving Nonword-to-Image Generation

## ABSTRACT

Text-to-image diffusion models sometimes depict blended concepts in generated images. One promising use case of this effect would be the nonword-to-image generation task which attempts to generate images intuitively imaginable from a non-existing word (nonword). To realize nonword-to-image generation, an existing study focused on associating nonwords with similar-sounding words. Since each nonword can have multiple similar-sounding words, generating images containing their blended concepts would increase intuitiveness, facilitating creative activities and promoting computational psycholinguistics. Nevertheless, no existing study has quantitatively evaluated this effect in either diffusion models or the nonword-to-image generation paradigm. Therefore, this paper first analyzes the conceptual blending in one of the pretrained diffusion models called Stable Diffusion. The analysis reveals that a high percentage of generated images depict blended concepts when inputting an embedding interpolating between the text embeddings of two text prompts referring to different concepts. Next, this paper explores the best text embedding space conversion method of an existing nonword-to-image generation framework to ensure both the occurrence of conceptual blending and image generation quality. We compare the conventional direct prediction approach with the proposed method that combines *k*-nearest neighbor search and linear regression. Evaluation reveals that the enhanced accuracy of the embedding space conversion by the proposed method improves the image generation quality, while the emergence of conceptual blending could be attributed mainly to the specific dimensions of the high-dimensional text embedding space.

## CCS CONCEPTS

• **Computing methodologies** → *Computer vision*; • **Information systems** → **Multimedia information systems**; *Nearest-neighbor search*.

## KEYWORDS

Diffusion models, Text-to-image generation, Conceptual blending, Nonwords

**ACM Reference Format:**
Anonymous Author(s). 2024. Investigating Conceptual Blending of a Diffusion Model for Improving Nonword-to-Image Generation. In *Proceedings of 31st ACM International Conference on Multimedia, October 28–November 01, 2024, Melbourne, Australia (MM '24).* ACM, New York, NY, USA, 9 pages. https://doi.org/XXXXXXX.XXXXXXX

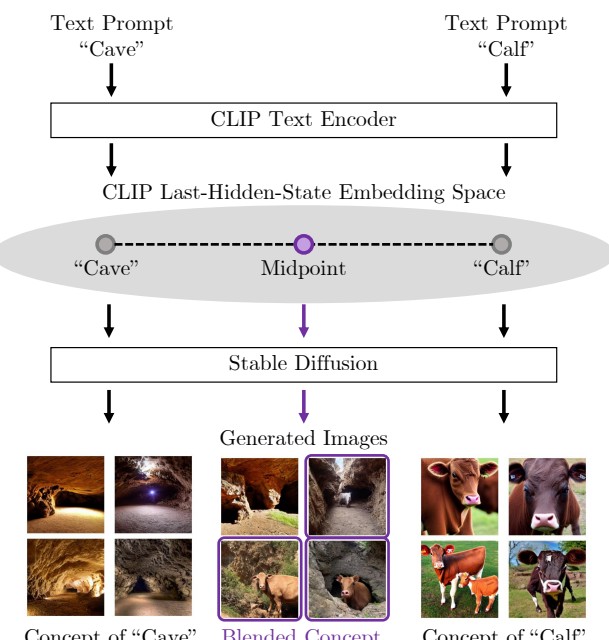

**Figure 1: Example of conceptual blending of a text-to-image diffusion model [18, 25] when generating images from an interpolated text embedding (midpoint) between the embeddings of two text prompts referring to different concepts [12].**

## 1 INTRODUCTION

Text-to-image diffusion models [18, 23] output generated images depicting blended concepts when an interpolated embedding between embeddings of multiple text prompts is input [12]. Figure 1 illustrates the conceptual blending exhibited by one of the diffusion models, Stable Diffusion [18, 25]. As it uses Contrastive Language-Image Pretraining (CLIP) [15] text encoder for computing conditioning text embeddings, it sometimes generates blended concepts (e.g., "*calf in a cave*") when inputting the midpoint of the CLIP text embeddings of two prompts referring to different concepts ("*calf*" and "*cave*").

This conceptual blending suggests various promising use cases, although the effect itself has not been well-studied. One such use case is the nonword-to-image generation task [9–11] which aims to generate images intuitively imageable from a given non-existing word (nonword). Generating such images for a nonword can facilitate creative activities including brand naming and computational

psycholinguistics. One approach for this task suggested by an existing study [9–11] is to generate images depicting concepts of similar-sounding words, assuming that humans associate nonwords with the meanings of such words. Here, conceptual blending can improve intuitiveness when nonwords are associated with multiple words. For instance, if a nonword "*calve*" (/ˈkæv/[1]) is associated with two similar-sounding words "*calf*" (/ˈkæf/) and "*cave*" (/ˈkeɪv/), it can be more intuitive to generate images depicting the blended concept of the two words than depicting only either concept.

Nonetheless, none of the existing literature provides a clear answer as to under which conditions diffusion models exhibit conceptual blending, and how it emerges in nonword-to-image generation results. Therefore, this paper conducts two evaluations to assess the occurrence of conceptual blending in each situation. The first one evaluates the capability of a pretrained Stable Diffusion to exhibit conceptual blending by detecting the presence of blended concepts in generated images. Using these detection metrics, the second one evaluates the occurrence of the effect in nonword-to-image generation results. Here, we explore the best text embedding space conversion method in the nonword-to-image generation framework adopting the same pretrained Stable Diffusion model. The conventional framework [9–11] took a direct approach to train a Multi-Layer Perceptron (MLP) to transfer embeddings between spaces. However, this yields large information loss, which could lead to inaccurate and poor-quality image generation and a reduced chance of conceptual blending. Alternatively, we propose a more accurate method by combining $k$-nearest neighbor search and linear regression. Our evaluation aims to investigate how the reduced loss affects conceptual blending and image generation quality.

Accordingly, this paper makes the following two contributions:

- We quantitatively evaluate a pretrained Stable Diffusion to discover under which conditions it exhibits conceptual blending given an interpolated embedding between two concepts.
- We explore the best text embedding space conversion method in the existing nonword-to-image generation framework to analyze factors that affect the emergence of conceptual blending as well as the image generation quality.

## 2 RELATED WORK

### 2.1 Text-to-Image Diffusion Models

A diffusion model [23] is one of the generative models used in most recent text-to-image generation methods [1, 14, 16, 20, 25]. In contrast to conventional generative models [4, 7], it is characterized by its step-by-step image generation procedure which gradually removes noise from a noisy image until a clear image is obtained. It generates images conditioned on a text prompt by utilizing the text embedding computed for the prompt in each denoising step. A latent diffusion model [18] is a more computationally efficient variant. It differs from diffusion models in that it performs the denoising procedure on the latent space instead of the image pixel space. Stable Diffusion [25] is one implementation of such a latent diffusion model which adopts the CLIP [15] text encoder as the

conditioning text embedding calculator and is trained using a large-scale dataset crawled from the Web called LAION-5B [22].

### 2.2 Conceptual Blending

Conceptual blending stems from cognitive linguistics, denoting a cognitive task to blend different concepts in minds to form a new concept inheriting their characteristics [3, 17]. In informatics, Melzi et al. [12] was the first to focus on it in diffusion models. Through a case study, they found that a pretrained Stable Diffusion exhibits conceptual blending when generating images from interpolated text embeddings between two concepts, even without additional training. Yet, there has been no other work on conceptual blending in diffusion models and it is still unclear under which conditions these models blend concepts.

One reason would be its seemingly limited use cases. The general text-to-image generation paradigm assumes that users can give clear instructions as text prompts into the model. Hence, if users demand images depicting blended concepts, they can instruct the model by typing detailed texts (e.g., "an image blending both a calf and a cave"). However, some applications cannot require users to put such a detailed text prompt, making it hard to satisfy users' needs for blending concepts.

One such case is the nonword-to-image generation task [9–11] which will be described in Section 2.3, where the input contains primarily non-existing words (nonwords). Before improving the nonword-to-image generation performance, Section 3 of this paper quantitatively studies the emergence of the conceptual blending targeting Stable Diffusion.

### 2.3 Nonword-to-Image Generation Task

The nonword-to-image generation task attempts to generate images intuitively imageable from a given non-existing word (nonword) [9–11]. The difference to the general text-to-image generation task is that the nonword-to-image generation task has no explicit ground-truth concepts that must be depicted in generated images since nonwords have no general interpretation of their meanings. However, as psycholinguistic studies suggest [6, 8, 21], humans tend to associate specific meanings even with nonwords in a somewhat predictable way. Hence, generating intuitive images for nonwords can profit in various applications including brand naming and language learning, and can foster computational psycholinguistics.

One existing study [9–11] tackled this by focusing on the human nature of associating a nonword with its similar-sounding words and generating images containing the concepts of such words. They trained a nonword encoder called NonwordCLIP that computes CLIP embeddings for the spelling or pronunciation of nonwords considering their phonetically similar words and inserted it into the Stable Diffusion architecture in place of the CLIP text encoder.

To correct the domain gap between the CLIP embedding space output by their language encoder (pooled embedding space) and that required by Stable Diffusion (last-hidden-state embedding space), they trained an MLP to convert embeddings in the former space into the latter. However, such a direct approach yields large information loss because the CLIP pooled embedding space is a compressed space of the last-hidden-state embedding space

---

[1]This paper describes word pronunciation using International Phonetic Alphabet (IPA) symbols. These symbols are also used as input of the existing nonword-to-image generation method to calculate pronunciation similarity.

and thus less informative. This loss can affect the occurrence of conceptual blending and image generation quality.

Hence, Section 4 proposes a more accurate text embedding space conversion method. To bypass the information loss, the proposed method combines $k$-nearest neighbor search and linear regression. Our evaluation in Section 4 compares the proposed method with the conventional MLP-based approach in terms of both conceptual blending and image generation quality.

## 3 INVESTIGATING CONCEPTUAL BLENDING

This section quantitatively evaluates under which conditions conceptual blending emerges in Stable Diffusion. To assess this, we detect whether an image generation result exhibits conceptual blending by identifying the concepts depicted in each generated image.

### 3.1 Experimental Setup

*3.1.1 Task.* Given an interpolated embedding between two text prompts describing concepts A and B, respectively, we measure how often a text-to-image diffusion model exhibits conceptual blending of the two concepts. This paper selects Stable Diffusion[2] [25] as a diffusion model which uses the CLIP[3] [15] text encoder to compute the CLIP last-hidden-space text embeddings. For each interpolated embedding, $N$ images are generated for analysis.

*3.1.2 Evaluation Data.* This paper uses a list of 1,000 existing English nouns representing different concepts, referred to as *EvalNouns1000* in later sections. These nouns are randomly taken from the MRC Psycholinguistic Database [2] with the restrictions of word imageability and frequency. Low-imageable and low-frequent words are filtered out to ensure that the concepts can be depicted clearly in images and that the words are not rare in everyday use (See supplementary materials for more details). Next, for two nouns (denoting concepts A and B) selected from *EvalNouns1000*, an interpolated embedding in the CLIP last-hidden-state space, $\mathbf{e}^{\text{hidden}}$, is calculated as a linear interpolation between the text embeddings of the two nouns/concepts. In detail, for each pair of the embeddings $\mathbf{e}_{\text{A}}^{\text{hidden}}$ and $\mathbf{e}_{\text{B}}^{\text{hidden}}$ corresponding to concepts A and B, the interpolated embedding is calculated as

$$\mathbf{e}^{\text{hidden}} = r\mathbf{e}_{\text{A}}^{\text{hidden}} + (1 - r)\mathbf{e}_{\text{B}}^{\text{hidden}}, \tag{1}$$

where $r$ denotes an interpolation ratio ranging between 0 and 1. We use the prompt "a photo of a <WORD>" for calculating embeddings $\mathbf{e}_{\text{A}}^{\text{hidden}}$ and $\mathbf{e}_{\text{B}}^{\text{hidden}}$ where "<WORD>" denotes each concept. We create 1,000 such pairs by randomly choosing two nouns from *EvalNouns1000*. The interpolation ratio for each pair is randomly assigned from 0.1 to 0.9 with the step size 0.1. For each pair, $N$ images are generated using the interpolated embedding.

### 3.2 Detecting a Single Visual Concept

Conceptual blending can be regarded as image generation depicting multiple visual concepts. Hence, to detect this, we first need to classify whether each image depicts a single visual concept.

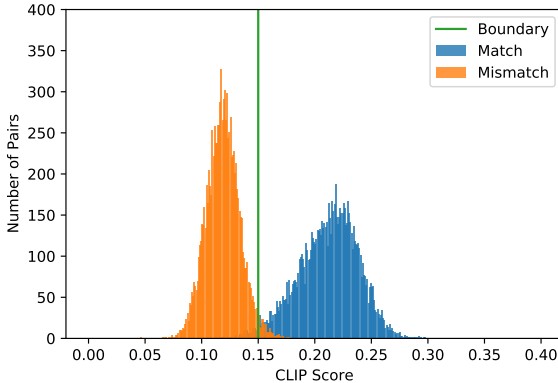

**Figure 2: Distribution of CLIP scores in matching and mismatching pairs and the classification boundary between the two classes.**

For this classification, we exploit CLIP score [15] which measures the cross-modal similarity between an image and a text. CLIP score is a cosine similarity between the embeddings of an image and a text encoded by a pretrained CLIP[4] [15]. A higher CLIP score between an image and a text indicates a higher likelihood of the image matching the text. Measuring this score between a generated image and a text that describes concept A enables detecting whether the single concept A is depicted in the image.

We approach this by finding the boundary on the CLIP score axis that best classifies the presence of a concept in a generated image. This paper defines it as the naïve Bayes decision boundary between the CLIP scores of matching and mismatching image-text pairs. The matching pairs are pairs of a generated image and its text prompt used for generating the image. The mismatching pairs are pairs of a generated image and a text prompt unrelated to the image generation. Here, 10,000 matching pairs are created by generating 10 images for each concept in *EvalNouns1000* with a prompt "a photo of a <WORD>". The mismatching pairs are created by shuffling the correspondence of the matching pairs. In calculating CLIP scores, we perform prompt engineering to increase the number of samples and thus the precision of the scores (See supplemental materials for more details). Figure 2 shows the distribution of the CLIP scores between each pair in the matching and mismatching pairs. The decision boundary 0.15 indicated in the figure classifies whether a given image-text pair matches or mismatches, detecting the presence of the concept denoted by the text in the image.

### 3.3 Detecting Conceptual Blending

This section detects whether a set of $N$ generated images for each interpolated embedding exhibits conceptual blending. We detect conceptual blending by identifying the depicted concepts in each image using the single-concept classifiers introduced in Section 3.2.

We first define two types of conceptual blending. Here, two image generation cases are distinguished involving conceptual blending between the two concepts A and B: *Blended Concept Depiction (BCD)*

---

[2]Stable Diffusion-v1-4 on the model card: https://github.com/CompVis/stable-diffusion/blob/main/Stable_Diffusion_v1_Model_Card.md (Accessed April 9, 2024)
[3]CLIP ViT-L/14 on the model card: https://github.com/openai/CLIP/blob/main/model-card.md (Accessed April 9, 2024)

[4]This paper uses CLIP ViT-L/14 for the calculation of CLIP score, too.

Concept A: "Armour"  Concept B: "Spider"  Blended Concept        Concept A: "Candle"  Concept B: "Pool"

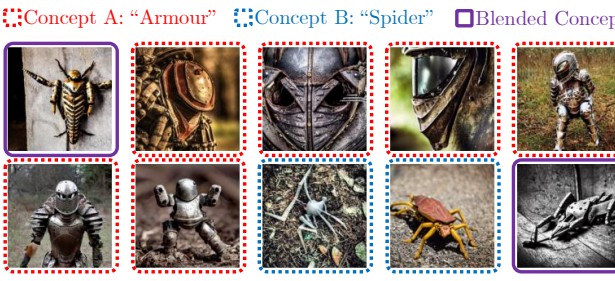
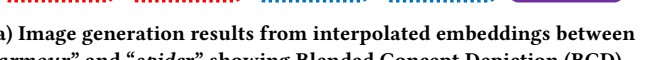
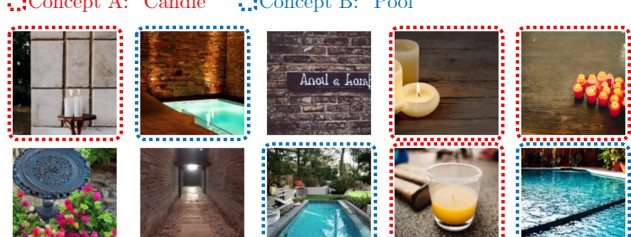

(a) Image generation results from interpolated embeddings between "*armour*" and "*spider*" showing Blended Concept Depiction (BCD).

(b) Image generation results from interpolated embeddings between "*candle*" and "*pool*" showing Mixed Concept Depiction (MCD).

**Figure 3: Examples of image generation results showing two types of conceptual blending targeted by this paper. Red, blue, and purple squares indicate cases where our method detected Concept A, Concept B, and both concepts, respectively. The images are generated from an interpolated embedding between Concepts A and B with an interpolation ratio of around 0.5.**

**Table 1: Ratios of respective cases when inputting interpolated embeddings between concepts A and B with different interpolation ratios. The number of pairs for each interpolation ratio used in the evaluation is also listed as a support.**

| Case | Interpolation Ratio of Concept A to Concept B | | | | | | | | | Total |
|---|---|---|---|---|---|---|---|---|---|---|
| | 0.1 | 0.2 | 0.3 | 0.4 | 0.5 | 0.6 | 0.7 | 0.8 | 0.9 | |
| Concept A | 0.152 | 0.311 | 0.411 | 0.709 | 0.948 | 0.981 | 1.000 | 1.000 | 1.000 | 0.732 |
| Concept B | 1.000 | 1.000 | 1.000 | 0.991 | 0.914 | 0.731 | 0.472 | 0.277 | 0.265 | 0.738 |
| BCD | 0.143 | 0.301 | 0.348 | 0.521 | **0.621** | 0.593 | 0.416 | 0.257 | 0.257 | 0.389 |
| MCD | 0.152 | 0.311 | 0.411 | 0.701 | **0.862** | 0.722 | 0.472 | 0.277 | 0.265 | 0.471 |
| Support | 105 | 103 | 112 | 117 | 116 | 108 | 125 | 101 | 113 | 1,000 |

and *Mixed Concept Depiction (MCD)*. BCD corresponds to the narrow sense of conceptual blending, where at least $n$ of the $N$ generated images show the blended concept of concepts A and B. MCD denotes a broader sense of conceptual blending, corresponding to the cases where at least $n$ images show concept A while at least $n$ images which do not necessarily show concept A, show concept B. We measure MCD as well as BCD because, although less direct than BCD, its emergence can still be a clue that both concepts A and B have been considered in an image generation result.

To detect both cases, we first detect the respective presence of concepts A and B in each of the $N$ images using the classifier introduced in Section 3.2. Then, we count BCD cases in which both concepts A and B are identified simultaneously in at least $n$ of the $N$ images. Meanwhile, we also count MCD cases where concepts A and B are identified in at least $n$ of the $N$ images, respectively. Figure 3 shows examples of BCD and MCD cases detected on image generation results.

### 3.4 Results

Table 1 shows the ratios of cases where Concept A, Concept B, BCD, and MCD are detected respectively in each image generation result. These are measured under the settings $n = 2$ and $N = 10$ (See supplemental materials for results under different settings).

We first observed a BCD ratio of 0.621 and an MCD ratio of 0.862 when inputting the midpoint between the embeddings of concepts A and B. This indicates that more than 60% and 85% of image generation results depicted blended and mixed concepts, respectively.

When aggregating all interpolation ratios, they dropped to 0.389 and 0.471, respectively. These ratios are still high because Stable Diffusion is not explicitly trained to visualize conceptual blending.

The results also revealed that the BCD and MCD ratios achieved the highest score at the interpolation ratio of 0.5. This aligns with the previous finding [12], suggesting that inputting the midpoint embedding maximizes the occurrence of conceptual blending.

Furthermore, we observed that the detected BCD cases contained two further subcases. One is the case where two concepts were depicted parallelly in an image (e.g., "*calf*" and "*cave*" shown in Fig. 1), and the other is where the textures of two concepts were blended (e.g., "*armour*" and "*spider*" shown in Fig. 3(a)). The former is more likely to occur when concepts A and B are often co-photographed in a real-world scene, whereas the latter is likely to occur when either of the concepts can grammatically work as an adjective. Thus, Stable Diffusion depicts "*calf*" and "*cave*" in parallel since "*calf*" can be in a "*cave*" in the real world (See Fig. 1). Meanwhile, since "*armour*" can work as the stem of an adjective as in "*armoured*", it can generate images depicting "*an armoured spider*".

## 4 NONWORD-TO-IMAGE GENERATION

Based on the findings reported in Section 3.4, this section investigates conceptual blending in nonword-to-image generation results.

### 4.1 Generalized Framework

Figure 4 shows a generalized framework proposed by an existing study [9–11] which utilizes a CLIP text encoder and Stable Diffusion to generate images for a nonword. First, a nonword encoder encodes a target nonword into the CLIP pooled embedding space to be in a location interpolating its similar-sounding words. To realize this, the existing study trains a language encoder NonwordCLIP by distilling the CLIP text encoder to project a nonword into the CLIP pooled embedding space. This distillation is performed in the pooled embedding space to ensure compatibility with the CLIP image encoder which also encodes images into the pooled embedding space. During this distillation, a phonetic prior is inserted into the nonword encoder to approximate nonword embeddings to those of the phonetically similar existing words.

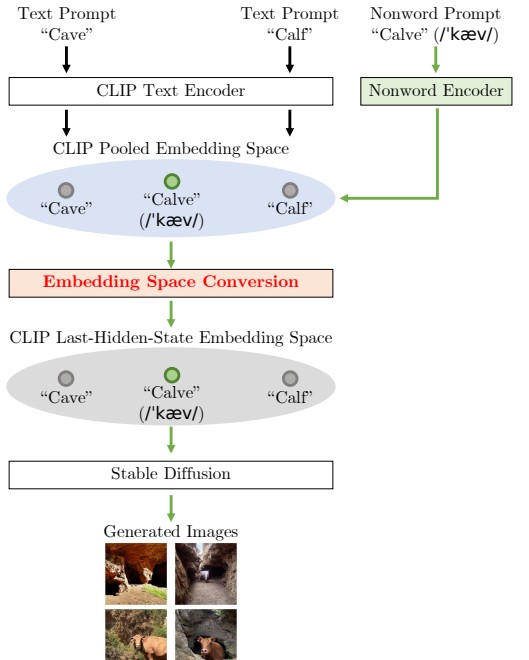

**Figure 4: Generalized framework for nonword-to-image generation [9–11]. The embedding space conversion method is improved to preserve the neighborhood relationships.**

After this nonword projection, an embedding space conversion method converts the pooled embedding into a corresponding last-hidden-state embedding before generating images using Stable Diffusion. This process is required by Stable Diffusion because it generates images from the CLIP last-hidden-state embedding space, not the pooled embedding space.

This paper adopts the same framework to first project nonwords into the pooled embedding space rather than projecting directly into the last-hidden-state embedding space, and then convert embeddings from the former space to the latter. We stick to this framework because using the CLIP pooled embedding space is a more common tradition than using other embedding spaces, and is also more flexible for extension to other image generation paradigms with different text-to-image generation models. For example, the same framework can be used for audio-to-image generation using a pretrained audio encoder distilled from the CLIP pooled embedding space [26] even if a generative model requires the CLIP penultimate layer's hidden-state embeddings, only with a minor adjustment.

### 4.2 Proposed Embedding Space Conversion

To convert embeddings between the two CLIP embedding spaces, the existing study [9–11] trained an MLP that reconstructs a last-hidden-state embedding from a pooled embedding. However, since the CLIP pooled embedding space is a compressed space of the last-hidden-state embedding space, it is impossible to perfectly

reconstruct last-hidden-state embeddings directly from pooled embeddings. This can lead to poor-quality image generation and a reduced chance of generating blended concepts.

To bypass this information loss, the proposed method takes a different approach; We combine nearest-neighbor search and linear regression to convert a pooled embedding into its last-hidden-state embedding. In the proposed method, last-hidden-state embeddings are calculated as interpolation in the last-hidden-state embedding space using the neighborhood relationships in the pooled embedding space. This approach improves the conversion accuracy because the last-hidden-state embedding is estimated not directly from the pooled embedding but based on the interpolation of the neighborhood embeddings in the last-hidden-state embedding space.

The proposed method first performs $k$-nearest neighbor search for a target nonword embedding $\mathbf{e}^{\text{pooled}}$ in the pooled embedding space, obtaining $k$ text embeddings $\mathbf{e}_1^{\text{pooled}}, \mathbf{e}_2^{\text{pooled}}, ..., \mathbf{e}_k^{\text{pooled}}$. Then, a linear regressor is trained to predict $\mathbf{e}^{\text{pooled}}$ from its $k$ nearest-neighbor embeddings, predicting $k$ optimized coefficients $\alpha_1, \alpha_2, ..., \alpha_k$, which minimizes the loss of

$$\mathbf{e}^{\text{pooled}} = \sum_{i=1}^{k} \alpha_i \mathbf{e}_i^{\text{pooled}}. \tag{2}$$

Lastly, we estimate the last-hidden-state embedding by performing a linear combination in the last-hidden-state embedding space using the optimized coefficients, which is formulated as

$$\mathbf{e}^{\text{hidden}} = \sum_{i=1}^{k} \alpha_i \mathbf{e}_i^{\text{hidden}}. \tag{3}$$

Our regressor does not employ a constant variable as the resulting intercept is valid only for regression in the pooled embedding space.

This estimation works only if the data distributions in two embedding spaces are similar. Our case should meet this requirement since the CLIP pooled embedding space is a space linearly compressed from the CLIP last-hidden-state embedding space [15].

### 4.3 Evaluating Embedding Space Conversion

This section evaluates the proposed embedding space conversion method in terms of information loss, neighborhood relationships, and image generation quality.

*4.3.1 Task.* Given an interpolated embedding in the pooled embedding space, the task is to estimate its last-hidden-state embedding that preserves the positional relationships with its neighbors with minimum loss. In this evaluation, we first prepare pairs of interpolated embeddings in both pooled and last-hidden-state embedding spaces. Equation (1) and its analogy to the pooled output space are used for calculating the interpolated embeddings in each space. For each pair, the last-hidden-state interpolated embedding is regarded as the ground truth for the pooled interpolated embedding. This evaluation does not use actual outputs of the nonword encoder as shown in the generalized framework because there is no straightforward way to prepare corresponding last-hidden-state embeddings.

*4.3.2 Implementation.* The proposed method performs $k$-nearest neighbor search on 26,143 data samples in the pooled embedding

space and uses the corresponding 26,143 samples for the estimation in the last-hidden-state embedding space. These samples, which we refer to as *TrainWords26143*, are taken from the existing study [9–11]. *TrainWords26143* consists of 26,143 words listed on the Spell Checker Oriented Word Lists (SCOWL)[5]. The previous work selected those words based on word frequency and pronunciation availability. In this paper, we create an embedding from each word using a prompt "a photo of a <WORD>". We also confirmed that our *EvalNouns1000* is a subset of *TrainWords26143*. This evaluation tests the hyperparameter $k$ with different integers ranging from 1 to 1,000.

We compare the proposed method with the MLP-based method identical to the one used in the existing study [9–11]. Their training data for the MLP used a set of 26,455 words. This set is almost the same as *TrainWords26143*, with the only difference being that it contains an additional 312 words for which no pronunciation was available. To increase samples, they created three prompts for each word in the wordlist: "<WORD>", "a photo of <WORD>", and "a photo of a <WORD>", although this augmentation is not performed in the proposed method.

*4.3.3 Evaluation Data and Metrics.* As evaluation data, we use the 1,000 matching pairs created in Section 3, which are pairs of two words randomly selected from *EvalNouns1000*, to prepare 1,000 interpolated embeddings.

As a metric for the information loss, we compute the L2 distance between a ground-truth embedding and an estimated embedding in the last-hidden-state embedding space averaged over all samples, which we call an L2 error. To see how well the neighborhood relationships are preserved, we also report Spearman's rank correlation between the rankings of neighborhood embeddings within *TrainWords26143* averaged over all samples. The rankings for ground-truth and estimated embeddings are obtained by searching their $\ell$ nearest-neighbor embeddings in the pooled and last-hidden-state embedding spaces, respectively. All the nearest-neighbor searches during this evaluation are performed on an L2 distance basis. The metric is measured under two different $\ell$s: 2 and 5 (See supplementary materials for results under more various settings). Before calculating these metrics, we flatten each last-hidden-state embedding shaped $77 \times 768$ into a $59,136$ dimensional vector.

Image generation quality is measured using Fréchet Inception Distance (FID) [5]. This evaluation measures two FIDs: $\text{FID}_{\text{Orig}}$ and $\text{FID}_{\text{Inter}}$. Calculating the former uses images generated from text embeddings of real text prompts which should depict clear visual concepts. In contrast, calculating the latter uses images generated from ground-truth interpolated embeddings which can exhibit conceptual blending as confirmed in Section 3. As a reference for calculating these metrics, 10 images are generated from each of the 1,000 text embeddings of *EvalNouns1000* and the 1,000 ground-truth interpolated embeddings introduced above[6] with a prompt.

*4.3.4 Results.* Results are shown in Table 2. We first observe that the L2 error of the proposed method was always lower than that of the comparative MLP-based method with a great margin. This demonstrates the strong advantage of our interpolation approach

[5]http://wordlist.aspell.net/ (Accessed April 9, 2024)
[6]We calculate FID using a Python package `pytorch-fid`: https://pypi.org/project/pytorch-fid/ (Accessed April 9, 2024)

**Table 2: Evaluation results for embedding space conversion methods. RCorr denotes Spearman's rank correlation.**

| Method | L2 Error ($\downarrow$) | RCorr$_{\ell=2}$ ($\uparrow$) | RCorr$_{\ell=5}$ ($\uparrow$) | FID$_{\text{Orig}}$ ($\downarrow$) | FID$_{\text{Inter}}$ ($\downarrow$) |
|---|---|---|---|---|---|
| MLP [9–11] | 245.38 | 0.846 | 0.783 | 16.03 | 11.59 |
| Ours ($k =$ 1) | 47.12 | **0.902** | 0.702 | 13.02 | 10.95 |
| Ours ($k =$ 2) | 37.38 | 0.880 | 0.788 | **12.25** | 7.58 |
| Ours ($k =$ 5) | 24.63 | 0.884 | 0.765 | 12.71 | 4.57 |
| Ours ($k =$ 10) | 19.29 | 0.882 | 0.771 | 13.21 | 3.52 |
| Ours ($k =$ 100) | 10.85 | 0.888 | 0.781 | 13.80 | 1.96 |
| Ours ($k =$ 200) | **10.30** | 0.886 | 0.791 | 13.85 | 1.87 |
| Ours ($k =$ 300) | 10.65 | 0.890 | 0.793 | 13.76 | **1.86** |
| Ours ($k =$ 400) | 11.68 | 0.890 | 0.797 | 13.66 | 1.95 |
| Ours ($k =$ 500) | 14.49 | 0.880 | **0.802** | 13.64 | 2.19 |
| Ours ($k =$ 1,000) | 39.14 | 0.882 | 0.799 | 13.73 | 4.11 |

over the direct prediction approach. The rank correlations of distances within a few neighborhood samples $\ell$ also showed a large gain over MLP, while tended to decrease as $\ell$ increased (See more results in supplementary materials). This can be explained as *the curse of dimensionality*, where the L2 distances measured in high-dimensional spaces become less diverse. Yet, the higher correlations of the proposed method within small $\ell$s indicate that it preserved neighborhood relationships better than the comparative method.

Furthermore, the proposed method showed better scores for both FID metrics than the comparative method. This indicates that our more precise and accurate last-hidden-state embedding estimation has improved the image generation quality, too. Notably, $\text{FID}_{\text{Inter}}$ showed 1.86 point at minimum when $k = 300$. This small value indicates that the generated images using the proposed method were almost identical to those generated using the ground-truth interpolated embeddings. Since we have confirmed in Section 3 that the ground-truth embeddings can yield BCD in up to 60% image generation results, it suggests that the proposed method with $k = 200$ or $k = 300$ can also yield conceptual blending in a similar frequency, which will be assessed more deeply in the next section.

## 4.4 Assessing Nonword-to-Image Generation

Lastly, we assess conceptual blending in image generation results generated for actual nonwords.

*4.4.1 Implementation.* As an encoder to compute CLIP pooled embeddings for nonwords, we retrain the NonwordCLIP-P [9–11] pronunciation encoder with a customized dataset. Since the original training dataset contained various sentences from image captioning datasets, the trained model tended to be biased on the word frequency in the dataset. For instance, in the case of the nonword "*calve*" (/ˈkæv/) which is assumed to have the two most similar-sounding words "*calf*" (/ˈkæf/) and "*cave*" (/ˈkeɪv/), its nonword embeddings were always encoded in a similar position to "*calf*" because the dataset contained "*calf*" more frequently than "*cave*".

To avoid this, we construct a dataset in which each word appears almost an equal number of times. The dataset consists of 5,496 highly-imageable and -frequent nouns and noun phrases created by combining the MRC Psycholinguistic Database [2], a Python package `wordfreq` [24], and an English lexical database WordNet [13] (See supplementary materials for more details). We augment the

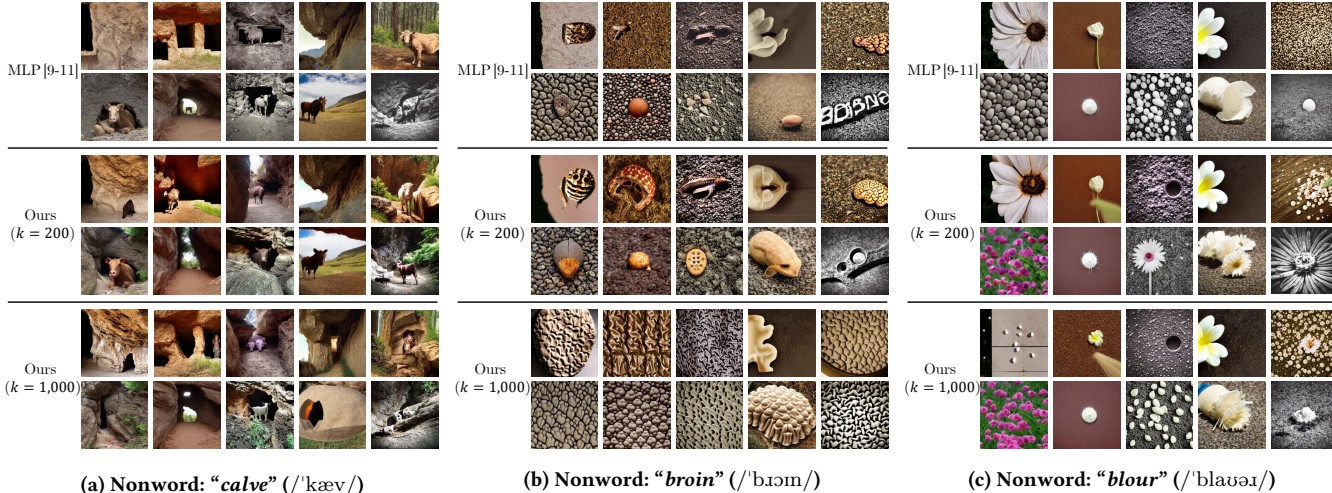

**(a) Nonword: "*calve*" (/ˈkæv/)**   **(b) Nonword: "*broin*" (/ˈbɹɔɪn/)**   **(c) Nonword: "*blour*" (/ˈblaʊəɹ/)**

**Figure 5: Nonword-to-image generation results exhibiting conceptual blending generated using different methods.**

dataset twice using the two prompts "<WORD>" and "a photo of a <WORD>", resulting in the training data of 10,992 samples.

When generating images, we follow the same settings as the exiting study [9–11], adopting the pronunciation prompt /ə ˈfoʊˌtoʊ ˈʌv ə <NONWORD>/ (corresponding to "a photo of a <NONWORD>").

*4.4.2 Evaluation Data and Metrics.* BCG and MCG ratios introduced in Section 3 are measured to assess conceptual blending in nonword-to-image generation results. To calculate these, this evaluation lacks ground-truth concepts A and B since the nonword embeddings are not computed by interpolating the embeddings of two concepts. Hence, to prepare pseudo-concepts A and B, we search the top-2 nearest-neighbor words for each nonword embedding in the CLIP pooled embedding space. For instance, if the top-2 nearest-neighbor embeddings of the nonword embedding "*calve*" are "*calf*" and "*cave*", our metrics detect occurrences of conceptual blending of "*calf*" and "*cave*" in the image generation result for "*calve*".

To align the metric calculation condition to Section 3, the nearest neighbors are searched from words in *EvalNouns1000*, and the hyperparameters for the metrics are set as $N = 10$ and $n = 2$. When searching the 2nd nearest-neighbor word, we exclude words that are too close to the 1st nearest-neighbor word from the candidates. This is to avoid concepts A and B being semantically too similar (e.g., "*bloom*" and "*blossom*") and mis-detecting the image generation of only concept A as the emergence of conceptual blending. The threshold to judge closeness is set as the 1st percentile of the distribution of the L2 distance between each of $_{1,000}C_2$ pairs of two samples in *EvalNouns1000*.

For nonwords, we use Sabbatino et al.'s 270 randomly created English nonwords [19]. This evaluation uses only 242 nonwords whose embeddings are not located in a close position to existing words. Specifically, we filter out nonwords to restrict them to be located in positions where the ratio of the distances to the top-1 and top-2 nearest-neighbor words, which corresponds to the interpolation ratio in Section 3, is between 0.4 and 0.6. 28 nonwords

**Table 3: Ratios of respective cases detected in generated images for 242 English nonwords.**

| Method | 1st-NN Concept | 2nd-NN Concept | BCD | MCD |
|---|---|---|---|---|
| MLP [9–11] | 0.930 | 0.847 | 0.669 | **0.789** |
| Ours ($k =$ 1) | 0.913 | 0.620 | 0.545 | 0.566 |
| Ours ($k =$ 2) | 0.913 | 0.669 | 0.550 | 0.607 |
| Ours ($k =$ 5) | 0.959 | 0.769 | 0.603 | 0.727 |
| Ours ($k =$ 10) | 0.926 | 0.798 | 0.579 | 0.727 |
| Ours ($k =$ 100) | 0.913 | 0.781 | 0.595 | 0.719 |
| Ours ($k =$ 200) | 0.938 | 0.810 | 0.624 | 0.760 |
| Ours ($k =$ 300) | 0.884 | 0.810 | 0.607 | 0.723 |
| Ours ($k =$ 400) | 0.884 | 0.818 | 0.612 | 0.740 |
| Ours ($k =$ 500) | 0.876 | 0.810 | 0.628 | 0.723 |
| Ours ($k =$ 1,000) | 0.913 | 0.868 | **0.707** | **0.789** |

are excluded because, as confirmed in Table 1, such embeddings are less likely to yield conceptual blending and can disturb metrics.

*4.4.3 Results.* The results are shown in Table 3. First, we confirmed that the proposed method yielded a maximum BCG ratio of 0.707 and an MCD ratio of 0.789 when $k$ was set to 1,000. This BCG ratio is much larger than the maximum value observed in Table 1 when the interpolation ratio was 0.5. The MCG and BCG ratios of the proposed method had another local maximum at $k = 200$, where the previous evaluation suggested the most accurate embedding conversion in the L2 error. These results indicate that the accurate embedding space conversion method did increase the chance of conceptual blending, but also suggest other factors that control the emergence of the effect. This can also be deduced by seeing the comparative method which yielded better BCD and MCD ratios than the proposed method with $k = 200$ while producing a larger L2 error in Table 2.

To seek insights into those factors, we next look at actual nonword-to-image generation results exhibiting conceptual blending. Figure 5 shows ten images for each of the three nonwords "*calve*" (/ˈkæv/), "*broin*" (/ˈbɹɔɪn/), and "*blour*" (/ˈblaʊəɹ/) generated using each

method. According to the top-2 nearest-neighbor words, the non-word "*calve*" has similar sounding words "*calf*" and "*cave*", "*broin*" has "*brain*" and "*bone*", and "*blour*" has "*flower*" and "*flour*". The figure indicates that all the methods can depict the blended concepts of the two similar-sounding words. Meanwhile, they mainly differed in image generation qualities and abstractness of the depicted objects, as indicated in the FID metrics in the previous evaluation. Most notably, the proposed method with $k = 1,000$ tended to exaggerate the texture of visual concepts, making the images very abstract. Also, as especially observable in Figs. 5(b) and 5(c), the MLP-based method tended to lack details of objects compared to the proposed method. Such abstractness can have overrated the concept detection metrics, as our classifier used in the metrics is prone to misdetection for images containing abstract objects that are recognizable in various ways.

As for why the inaccuracy of the embedding conversion did not affect conceptual blending, the dimensionality of the CLIP last-hidden-state embedding space can be the key. Last-hidden-state embeddings of the CLIP model used in this paper have the shape $77 \times 768$, where 77 denotes the maximum token length of a transformer model and each dimension corresponds to each token of the input text prompt. In our experimental setup, the input prompt used to create the evaluation data was always restricted to "a photo of a <WORD>". This prompt was generally tokenized into less than 10 tokens, suggesting that the dimensions of only the first less than 10 tokens were more essential than the other dimensions.

Considering this, we calculate the dimension-wise L2 error metric in the same setting as Table 2, as shown in Fig. 6. As expected, the MLP-based approach yielded more information loss than the proposed method in most dimensions. However, the loss became very comparable in the dimension of the first token corresponding to the [CLS] token, which usually represents the global feature of a last-hidden-state embedding. A similar trend can be seen in the position of the 7th token, where [EOS] (End Of Sentence) token typically falls. These results indicate that the embedding conversion accuracy in the dimension of [CLS] and [EOS] tokens would be the most responsible for the occurrence of conceptual blending in text-to-image diffusion models. Stable Diffusion could be referring dominantly to these two dimensions for determining concepts to blend in our experimental setup.

## 5 CONCLUSION

This paper first quantitatively analyzed under which conditions text-to-image diffusion models exhibit conceptual blending. We targeted an existing pretrained model called Stable Diffusion [18, 25], finding that it blends concepts in a high percentage of generated images when inputting an embedding interpolating between the text embeddings of two text prompts referring to different concepts. Next, this paper explored the best embedding space conversion method in the nonword-to-image generation framework [9–11] to analyze factors that affect conceptual blending and image generation quality. We compared the conventional direct prediction approach with the proposed method combining $k$-nearest neighbor search and linear regression. The evaluation confirmed that the embedding space conversion accuracy improved by the proposed method contributed to better image generation quality. The result

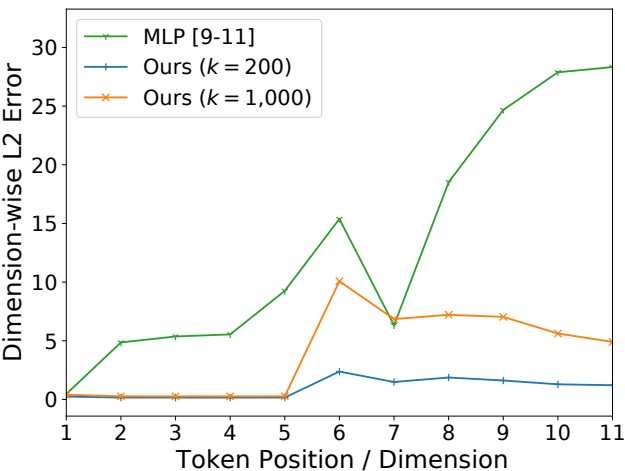

**Figure 6: L2 error for each dimension of the last-hidden-state embedding space corresponding to each token position up to the first 11 tokens.**

also suggested key dimensions in the high-dimensional text embedding space that could trigger the text-to-diffusion models to determine which concepts to blend.

As future work, investigating the correlation of the conceptual blending in diffusion models and nonword-to-image generation results with human cognition would be interesting. Furthermore, evaluations with diverse prompts could yield additional insights into conceptual blending in diffusion models.

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
