# OpenReview forum: "Investigating Conceptual Blending of a Diffusion Model for Improving Nonword-to-Image Generation"
_acmmm.org/ACMMM/2024/Conference — MM2024 Oral_

### Official Review · Reviewer_NQxs · 2024-05-16

**Rating:** 4
**Confidence:** 2

**Summary:**

In this work, the authors investigate the conceptual blending capability of a diffusion model. The study is two-fold. On the one hand, an experimental method to assess the occurrence of conceptual blending in generated images is devised. In particular, the authors propose a quantitative study to understand under which conditions this happens. On the other hand, the paper also analyzes the effects of conceptual blending on image generation quality. The authors develop a textual embedding space conversion method (based on nearest-neighbour search) to improve the generation results of a stable diffusion model when prompted with non-word embeddings. In line with previous literature, experiments reveal that conceptual blending is maximized when the interpolation ratio is around 0.5. Regarding image generation quality, the proposed embedding conversion method achieves higher quality generation compared to an MLP-based solution.

**Strengths:**

* This paper brings new insights to the community regarding the conceptual blending of diffusion models in the embedding space.

* Previous works investigated the blending property of a diffusion model mainly from a qualitative point of view. In this paper, the authors introduce an experimental approach to detect conceptual blending and to quantitatively assess under which conditions it emerges.

* The proposed embedding space conversion shows superior performance when compared to the MLP-based conversion, leading to higher image generation quality.

**Limitations:**

* In Section 3.3, the detection of conceptual blending is performed by using the single-concept classifier (with a 0.15 threshold tuned on the detection of single concepts). However, the authors do not demonstrate that the same threshold is still valid to classify blended concepts (when computing BCD and MCD).

* The paper lacks a discussion on the limitations of the proposed embedding space conversion technique. Which are the most important challenges? For example, which combination of nouns would lead to the worst image quality?
I believe that a discussion would increase the quality of the paper.

Additional questions (not influencing the final recommendation):

* I struggled to understand why in Table 2 FID\_inter is lower than FID\_orig. I would appreciate an explanation to interpret the results better.
* Why do the authors use Stable Diffusion 1.4 instead of version 2?

**Suitability:**

3

---

### Official Review · Reviewer_p3BH · 2024-05-25

**Rating:** 4
**Confidence:** 4

**Summary:**

The paper focusses on understanding the occurrence of concept blending in the text-to-image generation stable diffusion models. The authors also propose and embedding space conversion method based on nearest neighbours and linear regression to improve the conversion accuracy while going from the pooled to the last-hidden-last embedding in the existing nonword-to-image generation workflow itself.

**Strengths:**

The paper is well written and easy to follow, and the authors have discussed an interesting problem direction. The simple step of deciding the threshold for match vs mismatch based on the performed experiment is also interesting and intuitive, instead of randomly assigning a threshold value.

**Limitations:**

1. It is unclear how accurate the assumption of linearity while going from the pooled space to the last hidden state embedding is. Can the authors elaborate this aspect more?
2. Insufficient qualitative results - I would like to see more qualitative results for different input words
3. When the authors say that they "discover under which conditions the model exhibits conceptual blend" - it is unclear what these conditions refer to?

**Suitability:**

2

---

### Official Review · Reviewer_WpuM · 2024-05-28

**Rating:** 3
**Confidence:** 2

**Summary:**

This paper examines conceptual blending in the Stable Diffusion model, finding that many generated images blend concepts when interpolating text embeddings of different prompts. It also proposes a method combining 𝑘-nearest neighbor search and linear regression for better embedding space conversion, improving image quality and ensuring conceptual blending.

**Strengths:**

1. The topic of this paper is innovative, focusing on the problem of Nonword-to-Image Generation.

2. The paper provides a detailed introduction to the methods and experiments.

**Limitations:**

1. The novelty of the paper is limited. This paper is not proposing a completely new method on top of existing Nonword-to-Image Generation [9-11] but rather a simple and trivial method. Therefore, the novelty of the paper is worth concerned.

2. From an experimental standpoint, the paper does not provide a good comparison with previous methods. The author should specifically compare it with previous methods and point out the superiority of the existing methods.

3.The writing of the paper needs improvement. More images of the methods should be added to increase the clarity of the paper.

**Suitability:**

3

---

### Meta-Review · Area_Chair_nWL4 · 2024-07-04

**Recommendation:** Accept (Oral)
**Confidence:** 5

**Metareview:**

This paper investigates conceptual blending within the Stable Diffusion model, focusing on how generated images blend concepts when interpolating text embeddings. The authors propose a method combining 𝑘-nearest neighbor search and linear regression to improve embedding space conversion, thereby enhancing image quality and ensuring conceptual blending. The submission brings valuable insights into conceptual blending in diffusion models, presenting a method that, although simple, effectively improves embedding space conversion. The paper's limitations, particularly in terms of novelty and comparative analysis, should be addressed in future work. Overall, the solid experimental approach and the authors' responses to reviewers' concerns justify a final rating of Accept (Poster).